# Variational Graph Structure Learning for GNNs by using Marginal Likelihood

## Abstract

Learning graph structures for Graph Neural Networks (GNNs) can improve their performance, but it involves a challenging search over the large discrete space of all possible graphs. Prior works often enforce fixed constraints on the graph structure to induce properties such as sparsity, but such rigidity can be overly restrictive and harm performance. Here, we propose a simpler alternative to use the marginal likelihood which naturally favors such properties. We show that a variational formulation with Laplace's method automatically leads to a marginal likelihood based objective over discrete graph structures, which can be optimized efficiently using the Gumbel-Softmax trick. We call this approach the Laplace Approximation-based Graph Structure (LAGS) method, and show empirically that it improves the performance of different base GNNs, including recent state-of-the-art GNNs that outperform graph transformers.

## 1 Introduction

Graph neural networks (GNNs) (Gori et al., 2005; Scarselli et al., 2008; Kipf & Welling, 2017; Hamilton et al., 2017; Velickovic et al., 2018) rely on graph structure provided with the data, but such structures may not always truly represent the underlying dependencies. Often they are simply noisy and unreliable estimates of the true graph structure. Moreover, they can be incompatible with the chosen GNN architectures. For example, it has been shown that graphs with long-distance interactions or are heterophilic (dissimilar nodes tend to be connected) tend to perform poorly on most GNNs (Zhu et al., 2020; Alon & Yahav, 2021; Topping et al., 2022; NT & Maehara, 2019; Oono & Suzuki, 2019; Li et al., 2018). A common remedy is to change the architecture to better align with the data (Li et al., 2019; Chen et al., 2020a; Xu et al., 2018; Liu et al., 2020; Pei et al., 2020; Zhu et al., 2020), yet this still cannot resolve noisy and incorrect graph structures.

These problems can be avoided by learning the graph structure, but this is challenging because it involves searching over a large discrete space of all possible graphs. To reduce the cost, prior works often introduce explicit assumptions about the graph, enforced through rigid constraints. For example, some works induce global sparsity by penalizing the adjacency matrix (Luo et al., 2021; Jin et al., 2020), restricting edges to similar nodes (Fatemi et al., 2021; Liu et al., 2022; Gu et al., 2023), or limiting the graph to a predefined random graph family (Zhang et al., 2019; Ma et al., 2019; Lu et al., 2025; Yang et al., 2024). However, such rigid constraints risk conflicting with the unknown optimal graph structure for a given GNN architecture and task. Ideally, constraints should emerge automatically through regularization, guiding the search without relying on predefined assumptions about the structure. In this paper, we present such a method.

We propose using the marginal likelihood of the GNN as a simple solution for learning the graph structure. Marginal likelihood has long been used in Gaussian graphical models (Giudici & Green, 1999; Carvalho et al., 2007; Dobra et al., 2004), where the log-determinant of the covariance penalizes overly dense or heterophilic graphs (Dempster, 1972). Our main contribution is to extend this principle to GNNs via a variational formulation, where Laplace's method produces a marginal likelihood objective that automatically regularizes graph learning. The resulting objective is defined over the discrete space of graphs but can be efficiently optimized via the Gumbel Softmax trick. We refer to this approach as the Laplace Approximation-based Graph Structure (LAGS) learning.

Our contributions are as follows: (1) we introduce LAGS, a simple, GNN-agnostic framework that leverages marginal likelihood for graph structure learning (Sec. 3); (2) we show significant perfor-

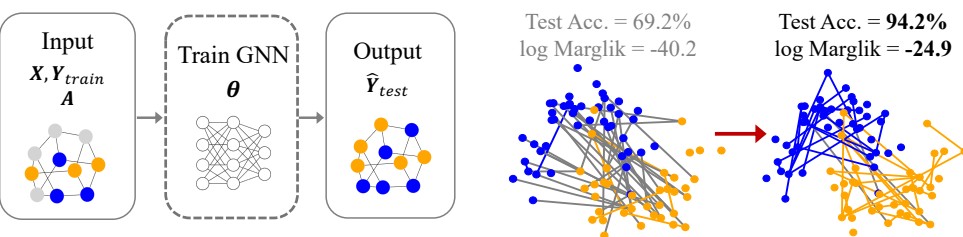

(a) GNN rely on graph $\mathbf{A}$, which may be noisy.    (b) Better graph has higher marginal likelihood.

Figure 1: Node classification with graph neural network (GNN). (a) GNN is trained on graph structure $\mathbf{A}$, so noise in $\mathbf{A}$ can lead to inaccurate predictions for unlabeled nodes (gray). (b) Marginal likelihood correlates with GNN performance. Learning graph structure by maximizing it removes noisy edges (gray) *(left)* and produces a more refined structure *(right)*, resulting in better accuracy.

mance gains from LAGS, with up to 8% higher accuracy on heterophilic graphs and up to 1% over the recent state-of-the-art GCN (Luo et al., 2024), which already outperforms graph transformers (Sec. 4.1, 4.2); and (3) we demonstrate through ablations that marginal likelihood effectively regularizes graph learning and learns edge probabilities aligned with edge importance (Sec. 4.3, 4.4).

## 2    GRAPH STRUCTURE LEARNING FOR GNNS

Consider the node classification task on a graph with $N$ nodes. Each node $i$ is associated with a feature vector $\mathbf{x}_i \in \mathbb{R}^d$, and for nodes in the training set we additionally observe a one-hot label vector $\mathbf{y}_i \in \{0,1\}^C$, where $C$ is the number of classes. The observed (possibly noisy) graph structure is represented as a binary adjacency matrix $\mathbf{A}_0$ of size $N \times N$, where $(i,j)$'th entry of 1 denotes an edge between node $i$ and $j$, while 0 indicates its absence. We denote the nodes' data as $\mathcal{D} = (\mathbf{X}, \mathbf{Y})$, where the node features $\mathbf{X} \in \mathbb{R}^{N \times d}$ has $\mathbf{x}_i^\top$ as rows and node labels $\mathbf{Y} \in \{0,1\}^{N \times C}$ has $\mathbf{y}_i^\top$ as rows. Our objective is to jointly learn a refined graph $\mathbf{A}$ and GNN parameters $\boldsymbol{\theta}$ for predicting unlabeled node classes.

### 2.1    GRAPH NEURAL NETWORKS

Graph neural network (GNN) is a class of models that learns representations by aggregating and transforming the features using the graph structure to learn latent node representations, denoted by $\mathbf{Z}^{(l)} \in \mathbb{R}^{N \times d_l}$ for the output of the $l$'th layer with width $d_l$. Different aggregation and transformation functions lead to the various GNN architectures. Here we consider two GNN variants: Graph Convolutional Networks (GCN) (Kipf & Welling, 2017) and GraphSAGE (Hamilton et al., 2017), which continue to be powerful and widely used models for learning on graph (Luo et al., 2024; Wang et al., 2024; Dinh et al., 2025).

Graph Convolutional Networks (GCNs) combine aggregation and transformation in a single step: $\mathbf{Z}^{(l)} = \sigma(\hat{\mathbf{A}} \mathbf{Z}^{(l-1)} \boldsymbol{\Theta}^{(l)})$, where $\hat{\mathbf{A}} = \mathbf{D}^{-\frac{1}{2}} \tilde{\mathbf{A}} \mathbf{D}^{-\frac{1}{2}}$, $\mathbf{D}$ is the degree matrix of $\tilde{\mathbf{A}} = \mathbf{A} + \mathbf{I}_N$, $\boldsymbol{\Theta}^{(l)} \in \mathbb{R}^{d_{l-1} \times d_l}$ are the layer weights, $\sigma(\cdot)$ is a nonlinearity (e.g. ReLU), and $\mathbf{Z}^{(0)} = \mathbf{X}$. In contrast, GraphSAGE applies separate transformations to a node's own features and its neighbors. After aggregating neighbor features (e.g. by summation), the result is concatenated with the node's features before transformation: $\mathbf{Z}^{(l)} = \sigma(\boldsymbol{\Theta}^{(l)} \cdot \mathrm{Concat}(\mathbf{A} \mathbf{Z}^{(l-1)}, \mathbf{Z}^{(l-1)}))$, with $\boldsymbol{\Theta}^{(l)} \in \mathbb{R}^{2d_{l-1} \times d_l}$.

For node classification, predictions $\hat{\mathbf{Y}}$ are obtained by applying the softmax function in the final layer. The GNN parameters, denoted $\boldsymbol{\theta} = \{\boldsymbol{\Theta}^{(l)}\}_{l=1}^L$, are learned by minimizing a regularized empirical loss, whose dependence on the graph structure $\mathbf{A}$ we make explicit:

$$\bar{\ell}(\boldsymbol{\theta}; \mathcal{D}, \mathbf{A}) = \sum_{i=1}^N \ell(\boldsymbol{\theta}; \mathcal{D}, \mathbf{A}) + r(\boldsymbol{\theta}). \tag{1}$$

For classification, $\ell(\boldsymbol{\theta}; \mathcal{D}, \mathbf{A})$ is typically the cross-entropy loss between the true label and the prediction. The regularizer $r(\boldsymbol{\theta})$ may be explicit or implicit. We distinguish $\mathbf{A}$, the graph structure used

in the GNN, from $\mathbf{A}_0$, the observed graph structure, to highlight that $\mathbf{A}$ need not coincide with $\mathbf{A}_0$. In standard GNNs, the graph structure $\mathbf{A}$ is simply taken to be the observed graph $\mathbf{A}_0$.

## 2.2 LEARNING GRAPH STRUCTURES

Standard use of GNNs assumes that the observed graph structure $\mathbf{A}_0$ encodes meaningful relationships among the nodes and fixes the computation graph $\mathbf{A}$ to $\mathbf{A}_0$. However, this assumption is often problematic: $\mathbf{A}_0$ may be noisy or inaccurate, and even when it is not, it may still be misaligned with the chosen GNN architecture. Learning graph structure provides a remedy by preserving and augmenting only the most relevant relationships while avoiding the introduction of additional noise. Throughout, we denote the unknown graph structure by $\mathbf{A}$. The task of jointly learning $\mathbf{A}$ and the GNN parameters $\boldsymbol{\theta}$ is known as the graph structure learning (GSL) problem.

## 2.3 RELATED WORKS

**Graph structure learning (GSL).** The method to learn the graph topology jointly with the GNN parameters can be broadly categorized into attention-based and direct approaches. Attention-based methods learn a similarity metric to add or remove edges (Velickovic et al., 2018; 2020; Chen et al., 2020b; Huang et al., 2020; Wu et al., 2022; Gu et al., 2023; In et al., 2024; Zhao et al., 2023), while direct methods treat the graph structure itself as learnable parameters (Zhang et al., 2019; Franceschi et al., 2019; Hasanzadeh et al., 2020; Jin et al., 2020; Chen et al., 2020b; Wang et al., 2021; Zhao et al., 2023). Attention-based methods are parameter-efficient, but they construct edges by linking nodes with similar representations under a chosen metric. This both requires node representations to align with the graph structure and makes it difficult to incorporate structural priors (e.g. from the observed graph), since parameters are not defined directly on the adjacency. We therefore adopt a direct graph learning approach.

To reduce the large discrete space of possible graphs, many GSL methods bias the learning process with assumptions about the target graph, often enforced through custom loss functions. Common strategies include penalizing edges between distant nodes to promote smoothness (Yang et al., 2019; Chen et al., 2020b; Jin et al., 2020), minimizing the $l_0$ or $l_1$ norm of the adjacency matrix to encourage sparsity (Luo et al., 2021; Jin et al., 2020), or imposing low-rank constraints to promote homophily (Luo et al., 2021). However, such rigid constraints, especially when applied uniformly across the graph, can be overly restrictive. In attention-based methods, sparsity is often enforced by restricting edges to the $k$-nearest neighbors or $\epsilon$-distance neighbors according to similarity or attention scores (Chen et al., 2020b; Fatemi et al., 2021; Liu et al., 2022; In et al., 2024). Another approach is to constrain the search to a predefined random graph family, often used in variational GSL methods.

**Variational GSL.** Rather than learning a single graph structure, variational GSL models a distribution over graphs. This is often constrained to a random graph family, such as the stochastic block model (Zhang et al., 2019; Lu et al., 2025; Yang et al., 2024; Lu et al., 2025), although performance heavily depends on the chosen family. Alternative approaches learn the graph posterior through mechanisms such as node-copying (Pal et al., 2019; Komanduri & Zhan, 2021), pairwise distance matrices (Pal et al., 2020), masking (akin to MC dropout) (Hasanzadeh et al., 2020; Lu et al., 2025), or Bernoulli parameter learning (Pantelis et al., 2020; Franceschi et al., 2019; Zhao et al., 2023). These methods either neglect the posterior over GNN parameters or rely on approximate techniques with a Bayesian interpretation, such as Monte Carlo dropout (Gal & Ghahramani, 2016). In contrast, our method explicitly leverages the GNN posterior through the marginal likelihood.

**Marginal likelihood.** The marginal likelihood (or evidence) is the integral of the likelihood with respect to the prior over parameters. When the posterior is approximated by a Gaussian with mean $\boldsymbol{\mu}$ and covariance $\boldsymbol{\Sigma}$, the log marginal likelihood can be approximated with Laplace's method as:

$$\log p(\mathcal{D}|\alpha) \approx \log p(\mathcal{D}|\boldsymbol{\theta}, \alpha) + \frac{1}{2} \log |\boldsymbol{\Sigma}| + \frac{d}{2} \log(2\pi) + \log p(\boldsymbol{\theta}|\alpha),$$

with $d$ the dimension of $\boldsymbol{\theta}$, hyperparameters $\alpha$, and prior $p(\boldsymbol{\theta}|\alpha)$. The first term reflects data fit (likelihood), while the second term—the log-determinant of the covariance—captures the entropy of the posterior and quantifies model complexity (Dempster, 1972; Good, 1963). In Bayesian model

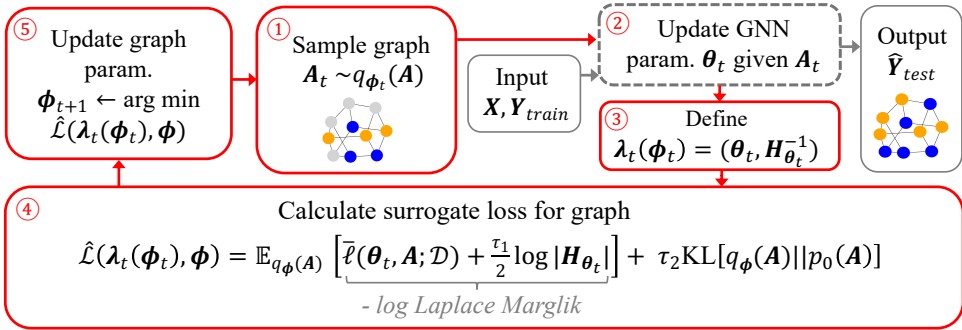

Figure 2: Overview of Laplace Approximation-based Graph Structure (LAGS) learning framework. The framework (red boxes) learns the graph structure during training of the GNN model.

selection, where model with higher marginal likelihood is preferred, the log-determinant term is the Occam factor that embodies Occam's razor (MacKay, 1992; Jeffreys, 1939; Gull, 1988). This allows marginal likelihood to automatically balance data fit and model complexity, favoring simpler models that explains the data well.

The determinant of the covariance also plays a central role in Gaussian graphical models (Giudici & Green, 1999; Carvalho et al., 2007; Dobra et al., 2004), where learning the graph reduces to covariance selection (Dempster, 1972) as the covariance matrix encodes the structure. Beyond fitting the data, it is desirable to obtain a minimal and sparse graph. Maximizing the determinant naturally induces such sparsity: a larger determinant corresponds to a sparser covariance matrix and thus a sparser graph. In our setting, the covariance is over GNN parameters rather than the graph itself, but the same effect holds–a larger determinant reflects a simpler model with more independent and flatter parameter distributions. When the graph structure simplifies the data, the model can also remain simple, which is reflected in a larger determinant. In contrast, dense and heterogeneous structures entangle features, demanding a more complex model and leading to a smaller determinant.

## 3 LAPLACE APPROXIMATION-BASED GRAPH STRUCTURE (LAGS)

We introduce the *Laplace Approximation-based Graph Structure (LAGS)* learning method, a GNN-agnostic framework that jointly learns GNN parameters and graph structure through marginal likelihood maximization. A variational formulation with Laplace's method yields the objective, which is optimized over discrete graphs using the Gumbel-Softmax trick. An overview of the method is shown in Fig. 2, with pseudo-code provided in Alg. 1.

### 3.1 VARIATIONAL GRAPH STRUCTURE LEARNING

We formulate learning of both the binary adjacency matrix $\mathbf{A} \in \{0,1\}^{N \times N}$ and GNN parameters $\boldsymbol{\theta}$ as a variational inference task. The joint posterior is approximated with a factorized variational distribution as follows:

$$q(\boldsymbol{\theta}, \mathbf{A}) = q(\boldsymbol{\theta})q(\mathbf{A}). \tag{2}$$

In contrast to conventional (non-variational) GNNs where $\boldsymbol{\theta}$ is learned as a point-estimate, here $\boldsymbol{\theta}$ is treated as a random variable under $q(\boldsymbol{\theta})$.

We assume that the posterior over $\boldsymbol{\theta}$ is a Gaussian with mean $\mathbf{m}$ and covariance $\boldsymbol{\Sigma}$:

$$q_{\boldsymbol{\lambda}}(\boldsymbol{\theta}) = \mathcal{N}(\boldsymbol{\theta}|\mathbf{m}, \boldsymbol{\Sigma}), \tag{3}$$

where we denote the parameter by $\boldsymbol{\lambda} = (\mathbf{m}, \boldsymbol{\Sigma})$. For the graph structure posterior $q(\mathbf{A})$, we assume it is a mean-field Bernoulli distribution over all $N^2$ potential edges, with $N$ being the number of nodes:

$$q_{\boldsymbol{\phi}}(\mathbf{A}) = \prod_{i=1}^{N} \prod_{j=1}^{N} \text{Ber}(a_{ij}|\pi_{ij} = \sigma(\phi_{ij})), \tag{4}$$

where $a_{ij} \in \{0, 1\}$ is the $(i, j)$'th entry of $\mathbf{A}$ and $\pi_{ij} \in (0, 1)$ is the probability that $a_{ij} = 1$ (an edge exists between node $i$ and $j$). To parameterize the edge probabilities $\pi_{ij}$ in an unconstrained way during optimization, we introduce real-valued logits $\phi_{ij} \in \mathbb{R}$ and set $\pi_{ij} = \sigma(\phi_{ij})$, where $\sigma(\cdot)$ is the sigmoid function. We denote $\phi$ as the vector of logits that parameterizes $q_\phi(\mathbf{A})$.

## 3.2 Joint learning of $\lambda$ and $\phi$

To learn the posterior parameters, we will minimize the following variational objective with temperatures $\tau_1, \tau_2 > 0$:

$$\mathcal{L}(\lambda, \phi) = \mathbb{E}_{q_{\lambda, \phi}(\theta, \mathbf{A})} \big[ \ell(\theta, \mathbf{A}; \mathcal{D}) \big] + \tau_1 \mathrm{KL} \big[ q_\lambda(\theta) || p_0(\theta) \big] + \tau_2 \mathrm{KL} \big[ q_\phi(\mathbf{A}) || p_0(\mathbf{A}) \big], \quad (5)$$

where $\ell(\theta, \mathbf{A}; \mathcal{D})$ is the unregularized empirical loss (from Eq. 1), and $p_0(\theta)$ and $p_0(\mathbf{A})$ are the prior distributions for $\theta$ and $\mathbf{A}$ respectively. We propose to perform a double-loop minimization of Eq. 5 where, at each iteration $t$ given $\phi_t$, we first minimize $\lambda$ using a cheap Laplace approximation to get $\lambda_t(\phi_t)$ in the inner loop, and then use this solution to find the next $\phi_{t+1}$. This is shown below,

$$\lambda_t(\phi_t) \leftarrow \arg\min_\lambda \mathcal{L}(\lambda, \phi_t), \qquad \phi_{t+1} \leftarrow \arg\min_\phi \widehat{\mathcal{L}}(\lambda(\phi_t), \phi), \quad (6)$$

where we denote the surrogate $\widehat{\mathcal{L}}$ used in the outer loop. The surrogate will utilize the marginal likelihood approximation, which is the main contribution of this work.

We start by discussing the minimization with respect to $\lambda = (\mathbf{m}, \boldsymbol{\Sigma})$ given $\phi_t$. We rely on the result by Khan & Rue (2023, App. C.1) who derive Laplace's method as a special case of variational learning, which we will use to simplify the training. Essentially, they show that approximating the expectation with respect to the Gaussian $q_\lambda(\theta)$ at its mean $\mathbf{m}$, via the so-called delta method, reduces variational learning to Laplace approximation. We provide detailed derivative in Appendix A and summarize them here. First, we approximate the expectation with respect to $q_{\phi_t}(\mathbf{A})$ at one sample, denoted as $\mathbf{A}_t \sim q_{\phi_t}(\mathbf{A})$, and expand the KL term, to obtain the following objective,

$$\begin{aligned} \mathcal{L}(\lambda, \phi_t) &\approx \mathbb{E}_{q_\lambda(\theta)} \big[ \ell(\theta, \mathbf{A}_t; \mathcal{D}) \big] + \tau_1 \mathrm{KL} \big[ q_\lambda(\theta) || p_0(\theta) \big] + \text{const.} \\ &= \mathbb{E}_{q_\lambda(\theta)} \big[ \bar{\ell}(\theta, \mathbf{A}_t; \mathcal{D}) \big] - \tau_1 \tfrac{1}{2} \log |\boldsymbol{\Sigma}| + \text{const.} \end{aligned} \quad (7)$$

This rewrites the problem in terms of the regularized empirical loss $\bar{\ell}(\theta, \mathbf{A}_t; \mathcal{D})$ from Eq. 1 where the regularizer is defined as $r(\theta) = -\tau_1 \log p_0(\theta)$. The objective can be further simplified to the Laplace approximation by using the second-order Taylor approximation at $\mathbf{m}$, giving us:

$$\mathcal{L}(\lambda, \phi_t) \approx \bar{\ell}(\mathbf{m}, \mathbf{A}_t; \mathcal{D}) - \tau_1 \tfrac{1}{2} \log |\boldsymbol{\Sigma}| + \mathrm{Tr} \big[ \mathbf{H_m} \boldsymbol{\Sigma} \big],$$

where $\mathbf{H_m} = \nabla^2 \bar{\ell}(\mathbf{m}, \mathbf{A}_t; \mathcal{D})$ is the Hessian at $\mathbf{m}$. Optimization with respect to $\mathbf{m}$ reduces to minimizing $\bar{\ell}(\theta, \mathbf{A}_t; \mathcal{D})$, whose minimizer is $\theta_t$. We therefore set $\mathbf{m}_t = \theta_t$. The optimization with respect to $\boldsymbol{\Sigma}$ is available in closed form by setting $\boldsymbol{\Sigma}_t = \nabla^2 \bar{\ell}(\theta_t, \mathbf{A}_t; \mathcal{D})^{-1}$, equivalent to $\mathbf{H}_{\theta_t}^{-1}$; details on Hessian approximations are given in Appendix A. By using the delta method, we have reduced the optimization over $\lambda$ to the standard Laplace approximation.

We now propose the surrogate loss used in the outer loop to optimize with respect to $\phi$ given $\lambda_t(\phi_t)$. Our main idea is to utilize the dependence of $\lambda_t(\phi_t)$ on the sample $\mathbf{A}_t \sim q_{\phi_t}(\mathbf{A})$, and replacing $\mathbf{A}_t$ with a free variable $\mathbf{A}$. We show this by first writing the objective (Eq. 5) at the solution $\lambda_t(\phi_t)$ and $\phi_t$, substituting $\mathbf{m}_t = \theta_t$ and $\boldsymbol{\Sigma}_t = \nabla^2 \bar{\ell}(\theta_t, \mathbf{A}_t; \mathcal{D})^{-1}$ to make the dependence on $\mathbf{A}_t$ explicit. In our proposed surrogate loss, $\mathbf{A}_t$ is replaced by a free variable $\mathbf{A}$ (shown in red):

Solution of Eq. 5 at $t$:

$$\mathcal{L}(\lambda_t(\phi_t), \phi_t) \approx \bar{\ell}(\theta_t, \mathbf{A}_t; \mathcal{D}) + \tau_1 \tfrac{1}{2} \log |\nabla^2 \bar{\ell}(\theta_t, \mathbf{A}_t; \mathcal{D})| + \tau_2 \mathrm{KL} \big[ q_{\phi_t}(\mathbf{A}) || p_0(\mathbf{A}) \big]$$

Proposed surrogate loss:

$$\widehat{\mathcal{L}}(\lambda_t(\phi_t), \phi) = \mathbb{E}_{q_\phi(\mathbf{A})} \big[ \bar{\ell}(\theta_t, \mathbf{A}; \mathcal{D}) + \tau_1 \tfrac{1}{2} \log |\nabla^2 \bar{\ell}(\theta_t, \mathbf{A}; \mathcal{D})| \big] + \tau_2 \mathrm{KL} \big[ q_\phi(\mathbf{A}) || p_0(\mathbf{A}) \big]. \quad (8)$$

The expectation term above is simply the negative log marginal likelihood obtained by the Laplace approximation at $\theta_t$ but the graph adjacency is now treated as a free variable $\mathbf{A}$ that can be optimized.

The advantage of this objective is that it incorporates the Hessian at $\theta_t$ that automatically regularizes the graph learning process. The log-determinant of the Hessian is the negative of the log-determinant

---

**Algorithm 1** Laplace Approximation-based Graph Structure (LAGS)

---

1: **Input:** training data $\mathcal{D}$, GNN learning rate $\eta$, graph learning rate $\gamma$, graph prior $p_0(\mathbf{A})$, temperatures $(\tau_1, \tau_2)$, burn-in epoch $B$, update freq $F$, steps $K$
2: $\boldsymbol{\phi}_0 \leftarrow$ initialize such that $\text{KL}[q_{\boldsymbol{\phi}_0}(\mathbf{A})||p_0(\mathbf{A})] = 0$
3: $\boldsymbol{\theta}_0 \leftarrow$ initialize GNN parameters
4: **for** each epoch **do**
5: $\quad \boldsymbol{\theta}_t \leftarrow$ optimize for $\bar{\ell}(\boldsymbol{\theta}, \mathbf{A}_t; \mathcal{D})$ (Eq. 1) with $\mathbf{A}_t \sim q_{\boldsymbol{\phi}_t}(\mathbf{A})$
6: $\quad$ **for** $k = 1, ..., K$ **every** $F$ epochs after burn-in $B$ **do**
7: $\quad\quad \widehat{\mathcal{L}}(\boldsymbol{\lambda}(\boldsymbol{\phi}_t), \boldsymbol{\phi}) \leftarrow$ Laplace approx. marginal likelihood surrogate loss (Eq. 8)
8: $\quad\quad \mathbf{g}_\phi \leftarrow$ gradient of $\widehat{\mathcal{L}}(\boldsymbol{\lambda}(\boldsymbol{\phi}_t), \boldsymbol{\phi})$ w.r.t. $\boldsymbol{\phi}$ calculated with Gumbel-Softmax trick
9: $\quad\quad \boldsymbol{\phi}_{t+1} \leftarrow$ optimizer step using normalized $\mathbf{g}_\phi$
10: $\quad$ **end for**
11: **end for**
12: Return $q_{\boldsymbol{\lambda}}(\boldsymbol{\theta}), q_{\boldsymbol{\phi}}(\mathbf{A})$

---

of the covariance, so minimizing Eq. 8 effectively maximizes the covariance determinant. As discussed in Sec. 2.3, this determinant reflects the posterior's entropy and provides a measure of model complexity (Dempster, 1972; Good, 1963). A larger value indicates a simpler model with higher posterior entropy, where parameter distributions are flatter and more independent. This, in turn, favors graph structures that simplify the data and can be explained by simple models, while discouraging dense or heterophilic structures that entangle class representations and require more complex models. The necessity of including the Hessian is empirically validated in Sec. 4.3, where removing it reduces the objective to likelihood maximization and results in degraded performance.

## 3.3 LEARNING DISCRETE GRAPH STRUCTURES

Since $\mathbf{A}$ is dicrete, standard gradient-based optimization methods cannot be used to update $\boldsymbol{\phi}$. One approach is to use the score function (REINFORCE) estimator which transforms the gradient of the expectation into the expectation of the gradient using the log-derivative trick, but such approach is known to have high variance. We instead propose to use the Gumbel-Softmax trick, which uses the concrete distribution to relax discrete random variables. The relaxed variable of $a_{ij}$ (corresponding to the edge between node $i$ and $j$) is denoted as $a_{ij}^r \in (0, 1)$, with probability distribution given by the binary Concrete distribution (from Maddison et al. (2017, App. B)):

$$a_{ij}^r = \frac{1}{1 + \exp(-\frac{\log \alpha_{ij} + L}{\tau})}, \qquad p(a_{ij}^r(\pi_{ij})) = \frac{\tau\alpha(a_{ij}^r(\pi_{ij}))^{-\tau-1}(1 - (a_{ij}^r(\pi_{ij})))^{-\tau-1}}{(\alpha(a_{ij}^r(\pi_{ij}))^{-\tau}(1 - (a_{ij}^r(\pi_{ij})))^{-\tau})^2}. \quad (9)$$

where $\alpha_{ij} = \frac{\pi_{ij}}{1 - \pi_{ij}}$ and $L = \log\frac{\epsilon_{ij}}{1 - \epsilon_{ij}}$ is a sample from the Logistic distribution, with $\epsilon_{ij} \sim \mathcal{U}(0, 1)$. The temperature $\tau$ controls the degree of relaxation. To make the relaxed $a_{ij}^r$ fully discrete, we use straight-through estimator (Bengio et al., 2013) which thresholds $a_{ij}^r$ to make it binary in the forward pass, while the gradient is calculated using the continuous $a_{ij}^r$. In the forward pass, the graph $\mathbf{A}$ is sampled from $q_{\boldsymbol{\phi}}(\mathbf{A})$ by sampling each edge $a_{ij}$.

## 4 EXPERIMENTS

We evaluate LAGS on node classification with two widely used GNN architectures: Graph Convolutional Networks (GCN) (Kipf & Welling, 2017) and GraphSAGE (Hamilton et al., 2017), which remain standard models for graph learning (Luo et al., 2024; Wang et al., 2024; Dinh et al., 2025). Our experiments examine: (1) the performance gains LAGS brings to base GNNs (Sec. 4.1); (2) the improvement of LAGS on state-of-the-art GNNs (Sec. 4.2); (3) the advantage of marginal likelihood in regularizing graph learning (Sec. 4.3); and (4) LAGS's ability to learn edge distribution that distinguish informative from noisy edges (Sec. 4.4). The implementation is provided in the supplementary material and will be released publicly upon acceptance. Experimental details and hyperparameters are given in Appendix C.

**Datasets.** We evaluate on standard benchmarks spanning homophilic and heterophilic networks. Citation graphs (Cora, Citeseer, Pubmed) use splits from Yang et al. (2016); Kipf & Welling (2017) (Table 1) and Luo et al. (2024) (Table 2). Social networks (BlogCatalog, Flickr) follow Huang et al. (2017); Zhao et al. (2021), and Wikipedia graphs (Squirrel, Chameleon, Roman-empire) follow Platonov et al. (2023). For details, see Appendix C.1.

## 4.1 PERFORMANCE GAINS FROM LAGS

Table 1: Node classification results (%) using standard hyperparameters. "OOM" and "OOT" denote out of memory and out of time (within 12 hours), respectively.

| Dataset | Cora | Citeseer | Pubmed | BlogCatalog | Flickr | Roman Empire |
|---|---|---|---|---|---|---|
| GCN | $81.31_{\pm0.46}$ | $71.19_{\pm0.51}$ | $78.62_{\pm0.42}$ | $75.48_{\pm0.37}$ | $63.71_{\pm0.37}$ | $52.52_{\pm0.38}$ |
| +LAGS | ↑1%$\mathbf{82.63}_{\pm0.24}$ | ↑1%$\mathbf{72.44}_{\pm1.54}$ | ↑.3%$\mathbf{78.88}_{\pm0.29}$ | ↑9%$\mathbf{84.93}_{\pm0.58}$ | ↑10%$\mathbf{73.91}_{\pm0.30}$ | ↑8%$\mathbf{60.72}_{\pm0.42}$ |
| GraphSAGE | $80.52_{\pm0.43}$ | $70.57_{\pm0.83}$ | $78.31_{\pm0.31}$ | $91.45_{\pm0.28}$ | $78.06_{\pm0.90}$ | $77.16_{\pm0.68}$ |
| +LAGS | ↑1%$\mathbf{81.10}_{\pm0.57}$ | ↑.7%$\mathbf{71.31}_{\pm0.69}$ | ↑.3%$\mathbf{78.64}_{\pm0.25}$ | ↑.2%$\mathbf{91.66}_{\pm0.30}$ | ↑1%$\mathbf{79.19}_{\pm0.62}$ | ↑1%$\mathbf{78.46}_{\pm0.58}$ |
| LDS | $84.24_{\pm0.40}$ | $74.79_{\pm0.42}$ | OOM | OOT | OOM | OOM |
| IDGL | $83.41_{\pm1.22}$ | $72.26_{\pm1.09}$ | OOM | $88.63_{\pm0.08}$ | $86.03_{\pm0.26}$ | OOM |
| IDGL-Anch | $81.14_{\pm0.67}$ | $70.62_{\pm0.99}$ | $81.56_{\pm1.25}$ | $85.00_{\pm0.59}$ | $69.83_{\pm1.11}$ | $33.15_{\pm0.97}$ |
| NodeFormer | $79.59_{\pm0.83}$ | $70.78_{\pm1.61}$ | $78.99_{\pm1.53}$ | $43.91_{\pm22.07}$ | $17.11_{\pm0.67}$ | $49.19_{\pm4.95}$ |
| WSGNN | $83.62_{\pm0.53}$ | $73.04_{\pm0.74}$ | OOM | $91.86_{\pm0.56}$ | $80.76_{\pm2.51}$ | OOM |
| SUBLIME | $77.90_{\pm1.14}$ | $73.10_{\pm0.54}$ | $80.76_{\pm0.69}$ | $88.90_{\pm0.87}$ | $86.88_{\pm0.46}$ | $60.95_{\pm0.30}$ |
| +unGSL | $79.20_{\pm1.38}$ | $71.13_{\pm1.37}$ | $78.10_{\pm0.92}$ | $92.14_{\pm0.85}$ | $86.80_{\pm0.20}$ | $62.45_{\pm0.38}$ |
| GraphGLOW | $82.12_{\pm0.85}$ | $73.44_{\pm0.69}$ | $79.56_{\pm0.42}$ | $26.04_{\pm1.57}$ | $18.14_{\pm0.47}$ | $39.19_{\pm0.71}$ |

**LAGS improves GNN performance.** We examine the performance gain of using LAGS with GCN and GraphSAGE, with results shown in Table 1. For both models, LAGS consistently improve upon the base model, with the largest gains on heterophilic graphs (BlogCatalog, Flickr, Roman-Empire), where LAGS-GCN improves accuracy by about 8%. Even on homophilic benchmarks (Cora, Citeseer, Pubmed), LAGS still produce measurable gains. Statistical significance is confirmed by p-values in Appendix C.5, and an ablation on graph priors is given in Appendix D.1. Notably, GCN benefits more from LAGS than GraphSAGE, likely because GCN is more sensitive to graph structure: its one-step convolution directly mixes node features with their neighbors, making it more vulnerable to suboptimal graphs (Zhu et al., 2020) and thus more responsive to structure learning.

**LAGS is competitive with other GSL methods.** We evaluate how LAGS position itself with other state-of-the-art graph structure learning algorithms (Zhou et al., 2023). Results in Table 1 show that LAGS's GCN and GraphSAGE variants are competitive with these methods, despite its simplicity and generality. While LDS (Franceschi et al., 2019), IDGL (Chen et al., 2020b) and WSGNN (Lao et al., 2022) achieve higher accuracy, they face scalability issues on larger datasets. IDGL-Anch, although more scalable, underperforms both variants of LAGS. Similarly, NodeFormer (Wu et al., 2022), despite being a transformer-based model, is outperformed by LAGS. SUBLIME (Liu et al., 2022) and unGSL (Han et al., 2025) are competitive but do not consistently surpass LAGS, whereas GraphGLOW (Zhao et al., 2023) struggles on heterophilic graphs. Moreover, compared to LAGS, these methods rely on more complex architectures: NodeFormer requires a transformer backbone, WSGNN trains two GNNs, SUBLIME adds an auxiliary learner, unGSL depends on another GSL method (e.g. SUBLIME), GraphGLOW relies on transfer learning from other graph data, and LDS requires validation data for graph learning. See Appendix C.7 for additional details.

**LAGS can be scalable.** We show that LAGS can be scaled by batching and applying a sparse prior on the graph. Directly learning parameters for all $N^2$ edges can be intractable, so we restrict learning to candidate edges from a k-nearest neighbor (kNN) graph and the observed graph. On ogbn-arxiv (Hu et al., 2020) (169k nodes, 1.16M edges), we demonstrate this using GraphSAGE, which naturally supports batching. With a kNN graph ($k = 5$) combined with the observed graph, LAGS improves GraphSAGE performance from $62.51\%$ ($\pm0.21$) to $63.46\%$ ($\pm0.06$). Training with a batch size of 1024 nodes required $\sim$4 hours and 38 GB of GPU memory, though both can be further tuned with sparser priors and different batch sizes. Additional details are provided in Appendix D.2.

## 4.2 IMPROVING UPON STATE-OF-THE-ART GNNS WITH LAGS

We evaluate the improvement LAGS can bring to state-of-the-art GNN. Recent work by Luo et al. (2024) shows that expanding the set of hyperparameters—such as batch normalization, layer normalization, and residual connections—can significantly improve the performance of classic GNNs (e.g. GCN), allowing them to surpass many state-of-the-art graph transformers. Following their setup, we adopt these enhancements for GCN (denoted as GCN*) and demonstrate that LAGS can further improve its performance. As shown in Table 2, LAGS provide additional boost to GCN*, and we include additional analysis of the learned graph in Appendix C.8 and p-values in Appendix C.5.

Table 2: Node classification results (%) using SoTA hyperparameters and setup from Luo et al. (2024). "*" denotes difference in model / data split from Table 1.

| Dataset | Cora* | Citeseer* | Pubmed* | Squirrel | Chameleon |
|---|---|---|---|---|---|
| GCN* | $84.21 \pm 0.96$ | $72.47 \pm 0.44$ | $80.10 \pm 0.76$ | $44.21 \pm 1.59$ | $44.35 \pm 4.30$ |
| +LAGS | ↑ .4% $\mathbf{84.61} \pm 0.74$ | ↑ .6% $\mathbf{73.10} \pm 0.48$ | ↑ .2% $\mathbf{80.33} \pm 0.84$ | ↑ .1% $\mathbf{44.64} \pm 2.02$ | ↑ 1% $\mathbf{45.32} \pm 4.17$ |

## 4.3 ABLATION ON MARGINAL LIKELIHOOD

To analyze the regularization effect of marginal likelihood in graph structure learning, we conduct an ablation comparing marginal likelihood (LAGS) with likelihood as the objective. The key distinction is the log-determinant of the Hessian term present in the marginal likelihood. As discussed in Sec. 3.2, this term quantifies the entropy of the GNN parameter posterior, which we hypothesize to be crucial for regulating the graph learning process. Removing this term reduces the surrogate loss (Eq. 8) to likelihood maximization. Results with GCN (Table 3) show that marginal likelihood yields consistently higher gains, while likelihood alone can even harm performance (as in Citeseer and Pubmed), confirming the regulating effect of the log-determinant term.

Table 3: Node classification accuracy (%) with GCN, comparing vanilla training, graph structure learning with likelihood objective, and with marginal likelihood (LAGS-GCN).

| Dataset | Cora | Citeseer | Pubmed | BlogCatalog | Flickr | Roman Empire |
|---|---|---|---|---|---|---|
| Vanilla | $81.31 \pm 0.46$ | $71.19 \pm 0.51$ | $78.62 \pm 0.42$ | $75.48 \pm 0.37$ | $63.71 \pm 0.37$ | $52.52 \pm 0.38$ |
| Likelihood | ↑ $81.98 \pm 0.17$ | ↓ $70.62 \pm 1.59$ | ↓ $78.06 \pm 0.67$ | ↑ $83.89 \pm 0.11$ | ↑ $72.38 \pm 0.67$ | ↑ $60.64 \pm 0.46$ |
| Marglik | ↑ $\mathbf{82.63} \pm 0.24$ | ↑ $\mathbf{72.44} \pm 1.54$ | ↑ $\mathbf{78.88} \pm 0.29$ | ↑ $\mathbf{84.93} \pm 0.58$ | ↑ $\mathbf{73.91} \pm 0.30$ | ↑ $\mathbf{60.72} \pm 0.42$ |

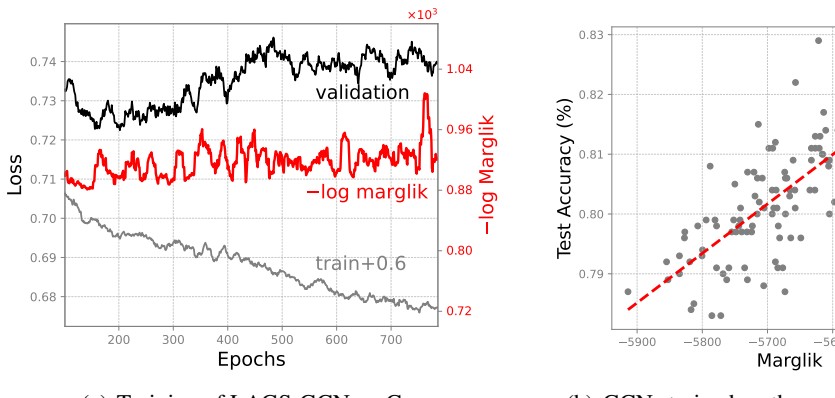

(a) Training of LAGS-GCN on Cora          (b) GCNs trained on the graph checkpoints

Figure 3: Marginal likelihood is a reliable proxy for GNN generalization and graph quality. (a) For LAGS-GCN on Cora, the log marginal likelihood matches the validation loss (MA window=15), providing a reliable measure of generalization. (b) Vanilla GCNs trained on checkpoint graphs show that better graph structures (with higher test accuracy) correlate with marginal likelihood.

We further show that marginal likelihood serves as a reliable proxy for both model generalization and the quality of the learned graph structure. As illustrated in Fig.3(a), unlike the training loss (likelihood), which decreases monotonically, the marginal likelihood closely tracks the validation loss. To isolate the effect of the graph structure, we train vanilla GCNs using checkpoints of the graphs learned during LAG-GCN training. As shown in Fig.3(b), graph structures with higher marginal likelihood consistently yield better test accuracy. These results confirm that marginal likelihood provides a principled objective for graph structure learning that aligns with model generalizability.

## 4.4 ABLATION ON LAGS'S LEARNED EDGE PROBABILITIES

We examine whether the posterior edge probabilities learned by LAGS capture edge importance. After training LAGS-GCN on Cora, we rank edges by their learned probabilities and then remove varying fractions of the highest and lowest probability edges to create perturbed graphs. These graphs are then used to train vanilla GCNs, and we evaluate how performance changes as high / low probability edges are dropped. The results in Fig. 4(a) show that removing high probability edges leads to a faster decline in performance faster than removing low probability edges. This indicates that LAGS is able to learn meaningful edge probabilities that correlate with their importance.

We repeat this with the attention learned using Graph Attention Network (GAT) (Velickovic et al., 2018) by dropping the highest and lowest attention weighted edges. Compared to GAT, we see that increasingly dropping the high probability edges (from LAGS) results in a faster drop in performance as opposed to dropping the high attention edges. This shows that the edge probabilities learned by LAGS correlate more strongly with edge importance than attention weights learned by GAT.

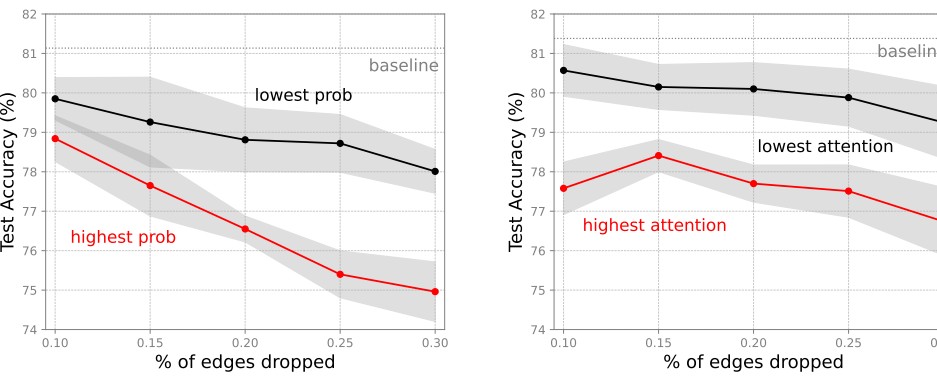

(a) Edge probabilities determined by LAGS-GCN      (b) Attention scores determined by GAT

Figure 4: LAGS learns edge probabilities that reflect informativeness. (a) Dropping high-probability edges from LAGS-GCN degrades performance more than low-probability ones. (b) GAT attention scores show weaker correlation, indicating LAGS captures edge importance more effectively.

## 5 CONCLUSION

We propose marginal likelihood as a simple and effective objective for graph structure learning in GNNs. Marginal likelihood naturally regularizes the structure learning process by favoring informative structures that simplify the data and can be explained by simple GNNs, while discouraging dense or heterophilic graphs that entangle node features and demand more complex models. Our framework, Laplace Approximation-based Graph Structure (LAGS), derives this objective via a variational formulation with Laplace's method and optimizes it over discrete graphs using the Gumbel-Softmax trick. Implemented with GCN and GraphSAGE, LAGS achieves up to 8% accuracy gains on heterophilic graphs and improves the state-of-the-art GCN* (Luo et al., 2024) by up to 1%. While computing Hessians may pose scalability challenges for large GNNs, efficient approximations offer promising directions, which we plan to explore in future work. We also aim to extend LAGS to attention-based graph learning methods as a potentially more scalable alternative.

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

## A DERIVATION OF THE LAPLACE APPROXIMATION

We provide a detailed derivation of the Laplace approximation for the optimization of $\boldsymbol{\lambda} = (\mathbf{m}, \boldsymbol{\Sigma})$, given $\boldsymbol{\phi}_t$. By simplifying the joint variational objective (Eq. 5) with $\boldsymbol{\phi}_t$, we obtain the variational objective for $\boldsymbol{\lambda}$ as:

$$\mathcal{L}(\boldsymbol{\lambda}, \boldsymbol{\phi}_t) = \mathbb{E}_{q_{\boldsymbol{\lambda}, \boldsymbol{\phi}_t}(\boldsymbol{\theta}, \mathbf{A})}\big[\ell(\boldsymbol{\theta}, \mathbf{A}; \mathcal{D})\big] + \tau_1 \mathrm{KL}\big[q_{\boldsymbol{\lambda}}(\boldsymbol{\theta}) || p_0(\boldsymbol{\theta})\big] + \text{const.}$$

We approximate the expectations with respect to $q_{\boldsymbol{\lambda}, \boldsymbol{\phi}_t}(\boldsymbol{\theta}, \mathbf{A})$ at a graph sample $\mathbf{A}_t \sim q_{\boldsymbol{\phi}_t}(\mathbf{A})$, and expand the KL term:

$$\mathcal{L}(\boldsymbol{\lambda}, \boldsymbol{\phi}_t) \approx \mathbb{E}_{q_{\boldsymbol{\lambda}}(\boldsymbol{\theta})}\big[\ell(\boldsymbol{\theta}, \mathbf{A}_t; \mathcal{D})\big] + \tau_1 \mathrm{KL}\big[q_{\boldsymbol{\lambda}}(\boldsymbol{\theta}) || p_0(\boldsymbol{\theta})\big] + \text{const.}$$
$$= \mathbb{E}_{q_{\boldsymbol{\lambda}}(\boldsymbol{\theta})}\big[\ell(\boldsymbol{\theta}, \mathbf{A}_t; \mathcal{D})\big] + \tau_1 \mathbb{E}_{q_{\boldsymbol{\lambda}}(\boldsymbol{\theta})}\big[\log q_{\boldsymbol{\lambda}}(\boldsymbol{\theta}) - \log p_0(\boldsymbol{\theta})\big] + \text{const.}$$
$$= \mathbb{E}_{q_{\boldsymbol{\lambda}}(\boldsymbol{\theta})}\big[\ell(\boldsymbol{\theta}, \mathbf{A}_t; \mathcal{D}) - \tau_1 \log p_0(\boldsymbol{\theta})\big] - \tau_1 \mathcal{H}(q_{\boldsymbol{\lambda}}) + \text{const.}$$
$$= \mathbb{E}_{q_{\boldsymbol{\lambda}}(\boldsymbol{\theta})}\big[\bar{\ell}(\boldsymbol{\theta}, \mathbf{A}_t; \mathcal{D})\big] - \tau_1 \tfrac{1}{2} \log |\boldsymbol{\Sigma}| + \text{const.}$$

Here, we denote by $\mathcal{H}(q)$ the entropy in the third line. The last line rewrite the problem in terms of the regularized empirical loss from Eq. 1 where the regularizer is defined as $r(\boldsymbol{\theta}) = -\tau_1 \log p_0(\boldsymbol{\theta})$. We can further simplify this objective to Laplace approximation by using second-order Taylor approximation at $\mathbf{m}$ and the delta method to remove the expectation. This is shown below by using the gradient and Hessian of $\bar{\ell}(\boldsymbol{\theta}, \mathbf{A}_t; \mathcal{D})$ at $\mathbf{m}$ denoted by $\mathbf{g_m}$ and $\mathbf{H_m}$ respectively,

$$\mathbb{E}_{q_{\boldsymbol{\lambda}}(\boldsymbol{\theta})}\big[\bar{\ell}(\mathbf{m}, \mathbf{A}_t; \mathcal{D}) + (\boldsymbol{\theta} - \mathbf{m})^\top \mathbf{g_m} + (\boldsymbol{\theta} - \mathbf{m})^\top \mathbf{H_m}(\boldsymbol{\theta} - \mathbf{m})\big] - \tau_1 \tfrac{1}{2} \log |\boldsymbol{\Sigma}|$$
$$\approx \bar{\ell}(\mathbf{m}, \mathbf{A}_t; \mathcal{D}) + \mathrm{Tr}\big[\mathbf{H_m}\boldsymbol{\Sigma}\big] - \tau_1 \tfrac{1}{2} \log |\boldsymbol{\Sigma}|.$$

The optimization with respect to $\mathbf{m}$ reduces to minimization of $\bar{\ell}(\boldsymbol{\theta}, \mathbf{A}_t; \mathcal{D})$. Denoting the minimizer by $\boldsymbol{\theta}_t$, we therefore set $\mathbf{m}_t = \boldsymbol{\theta}_t$. The optimization with respect to $\boldsymbol{\Sigma}$ is available in closed form by setting $\boldsymbol{\Sigma}_t = \mathbf{H}_{\boldsymbol{\theta}_t}^{-1}$. This reduces the optimization with respect to $\boldsymbol{\lambda}$ to the standard Laplace approximation.

## B APPROXIMATION OF THE HESSIAN

We discuss several approximations of the Hessian matrix $\mathbf{H}_{\boldsymbol{\theta}} = \nabla_{\boldsymbol{\theta}}^2 \bar{\ell}(\boldsymbol{\theta}, \mathbf{A}_t; \mathcal{D})$ used in the Laplace approximation. The Hessian is a $P \times P$ matrix, where $P$ is the number of parameters in the GNN model. For modern neural networks, $P$ can be in the order of millions, making direct computation and storage of the Hessian infeasible. We therefore consider several approximations to make the computation tractable. Our implementation of LAGS uses the Generalized Gauss-Newton (GGN) with block-diagonal approximation of the Hessian.

### B.1 GENERALIZED GAUSS-NEWTON (GGN)

Generalized Gauss-Newton (GGN) extends the Gauss-Newton matrix approximation to general loss function (Schraudolph, 2002; Martens, 2010) and approximates the Hessian w.r.t the parameters. The GGN matrix is given by:

$$\mathbf{H}_{\boldsymbol{\theta}} \approx \mathbf{H}_{\boldsymbol{\theta}}^{\mathrm{GGN}} = \sum_{i=1}^{N} [\mathbf{J}_{\boldsymbol{\theta}}^\top \mathbf{H}_L \mathbf{J}_{\boldsymbol{\theta}}]_i + \mathbf{P}_{\boldsymbol{\theta}},$$

where $\mathbf{J}_{\boldsymbol{\theta}} \in \mathbb{R}^{C \times P}$ is the Jacobian matrix w.r.t to the parameters, and $\mathbf{H}_L$ and $\mathbf{P}_{\boldsymbol{\theta}}$ are the Hessian of the negative log -likelihood and -prior, respectively. More specifically, $\mathbf{H}_L := -\nabla_{\mathbf{ff}}^2 \log p(\mathbf{y}|\mathbf{f})$ is the Hessian of the (cross-entropy) loss w.r.t to the predictions $\mathbf{f}$, therefore has dimension $C \times C$, while the Hessian of the negative log-prior $\mathbf{P}_{\boldsymbol{\theta}} := -\nabla_{\boldsymbol{\theta}}^2 \log p(\boldsymbol{\theta}|\mathcal{M})$ has dimension $P \times P$. So the resultant GGN matrix is $P \times P$ and the complexity is $O(NPC^2 + NCP^2)$.

### B.2 EMPIRICAL FISHER (EF)

The empirical Fisher (EF) (Martens & Grosse, 2015; Grosse & Salakhutdinov, 2015) estimates the Fisher information matrix using the data $\mathcal{D}$, and can be used to estimate the Hessian:

$$\mathbf{H}_{\boldsymbol{\theta}} \approx \mathbf{H}_{\boldsymbol{\theta}}^{\mathrm{EF}} = \sum_{i=1}^{N} \mathbf{g}_{\boldsymbol{\theta},i}^\top \mathbf{g}_{\boldsymbol{\theta},i} + \mathbf{P}_{\boldsymbol{\theta}},$$

where $\mathbf{g}_{\boldsymbol{\theta},i} \in \mathbb{R}^{P \times 1}$ is the gradient of the log-likelihood $\mathbf{g}_{\boldsymbol{\theta},i} := \nabla_{\boldsymbol{\theta}} \log p(\mathcal{D}_i|\boldsymbol{\theta}, \mathcal{M})$, and $\mathbf{g}_{\boldsymbol{\theta},i}^{\top} \mathbf{g}_{\boldsymbol{\theta},i} \in \mathbb{R}^{P \times P}$ is the outer product. The resultant EF matrix is $P \times P$ and the complexity is $O(NP^2)$. For various matching pairs (such as linear activation function and square error, sigmoid and cross-entropy, and softmax and negative log-likelihood), the Fisher information matrix and GGN are equivalent (Pascanu & Bengio, 2014); this further validates EF as an approximation for the Hessian (Bottou et al., 2018).

### B.3 BLOCK AND DIAGONAL APPROXIMATION

While the GGN and EF approximations allow for more manageable computation of the Hessian, practically storing and computing the determinant of a $P \times P$ matrix may still be infeasible. A solution is to make block diagonal or diagonal approximations of the Hessian. Since the Hessian is typically diagonal dominant, such an approximation is reasonable. Block diagonal approximation of the Hessian is calculated through Kronecker factorization of the Fisher matrix (Martens & Grosse, 2015) or the GGN matrix (Botev et al., 2017); that is, for each layer $l$ of the neural network: $[\mathbf{H}_{\boldsymbol{\theta}}]_l \approx \mathbb{E}[\mathbf{Q}_l] \otimes \mathbb{E}[\mathbf{G}_l]$, where $\mathbf{Q}_l \in \mathbb{R}^{d_{l-1} \times d_{l-1}}$ matrix is calculated from the input activations to the $l$-th layer, and $\mathbf{G}_l \in \mathbb{R}^{d_l \times d_l}$ is based on the gradient of the output ($d_l$ corresponds to the output dimension of the $l$-th layer). Diagonal approximation further reduces computation by only using the diagonal entries of the Hessian.

## C  EXPERIMENTS

We provide additional details on the experiments from Sec. 4, including statistics of datasets used, hyperparameters, experimental setup, and training statistics.

### C.1  DATASETS

We summarize the datasets used in our experiments in Table 4. Citation networks (Cora, Citeseer, Pubmed) are highly homophilic, where nodes of the same class are more likely to connect, while social networks (BlogCatalog, Flickr) and Wikipedia graphs (Squirrel, Chameleon, Roman-Empire) are more heterophilic. We follow the dataset splits of Yang et al. (2016); Kipf & Welling (2017); Platonov et al. (2023), except for Table 2, where we adopt the splits from Luo et al. (2024) for fair comparison.

Table 4: Overview of the datasets used in the experiments.

| **Dataset** | # Nodes | # Edges | # Features | # Classes | Homophily |
|---|---|---|---|---|---|
| Cora | 2,708 | 5,278 | 1,433 | 7 | 0.81 |
| Citeseer | 3,327 | 4,552 | 3,703 | 6 | 0.74 |
| Pubmed | 19,717 | 44,324 | 500 | 3 | 0.80 |
| BlogCatalog | 5,196 | 171,743 | 8,189 | 6 | 0.40 |
| Flickr | 7,575 | 239,738 | 12,047 | 9 | 0.24 |
| Squirrel | 2,223 | 93,996 | 2,089 | 5 | 0.21 |
| Chameleon | 890 | 17,708 | 2,325 | 5 | 0.24 |
| Roman-Empire | 22,662 | 32,927 | 300 | 18 | 0.05 |

### C.2  CODE

Our implementation for marginal likelihood optimization uses code from Immer et al. (2022), which uses the Automatic Second-order Differentiation Library (ASDL) from Osawa et al. (2023) for the Hessian approximation. We adapt them for our use case. Our code will be made publicly available upon acceptance. All experiments can be reproduced with a consumer grade GPU.

### C.3  OVERVIEW OF HYPERPARAMETERS

We provide additional details on the hyperparameters specific to LAGS.

**Graph learning rate** $\gamma$. The learning rate for $\phi$. While using a lower $\gamma$ with more steps $K$ allow for more stable updates, it comes at the expense of more computations.

**Graph prior probabilities** $\boldsymbol{\pi}_0$. The prior probabilities $\boldsymbol{\pi}_0 \in \mathbb{R}^{N \times N}$ where each entry $\boldsymbol{\pi}_{0_{i,j}} \in (0, 1)$ is the prior probability of edge $(i, j)$ being present in the graph. Defining $\boldsymbol{\pi}_0$ can be done in many different ways, such as using kNN-graph or using the observed graph, as we have done in our implementation.

In our implementation, we define the graph prior $\boldsymbol{\pi}_0$ by setting $\boldsymbol{\pi}_{0_{i,j}} = \pi^o$ if edge $(i, j)$ is in the observed graph $\mathbf{A}$, and $\boldsymbol{\pi}_{0_{i,j}} = \pi^u$ otherwise. We found that for GCN, for high quality homophilic graphs (e.g. Cora, Citeseer and Pubmed), it is best to set $\pi^o$ to a value close to 1. While for low quality or heterophilic graphs (e.g. BlogCatalog and Flickr), it is best to set $\pi^o$ to be close to 0. For GraphSAGE, however, we found that in both cases, it is best to set the observed edges to be close to 1. In the case that the kNN graph is used as the prior, we define $k$ as the number of nearest neighbors and $\pi^k \in (0, 1)$ as the probability of the kNN edges being present.

**Temperatures** $(\tau_1, \tau_2)$. The temperatures determine the strength of the KL-divergence terms of Eq. 5 for $\boldsymbol{\lambda}$ and $\phi$ respectively. To reiterate Eq. 5, the loss is:

$$\mathcal{L}(\boldsymbol{\lambda}, \boldsymbol{\phi}) = \mathbb{E}_{q_{\boldsymbol{\lambda},\boldsymbol{\phi}}(\boldsymbol{\theta}, \mathbf{A})}\big[\ell(\boldsymbol{\theta}, \mathbf{A}; \mathcal{D})\big] + \tau_1 \mathrm{KL}\big[q_{\boldsymbol{\lambda}}(\boldsymbol{\theta})\|p_0(\boldsymbol{\theta})\big] + \tau_2 \mathrm{KL}\big[q_{\boldsymbol{\phi}}(\mathbf{A})\|p_0(\mathbf{A})\big],$$

where the first term is the expectation of the likelihood. Temperature $\tau_1$ directly corresponds to the regularization of the parameters $\boldsymbol{\theta}$, while $\tau_2$ corresponds to the regularization of the graph posterior.

**Burn-in epoch** $B$. The number of epochs before the graph posterior is updated.

**Update frequency** $F$. The graph posterior is updated every $F$ epochs.

**Steps** $K$. The number of steps taken to update the graph posterior.

## C.4 EXPERIMENTAL SETUP AND HYPERPARAMETERS

The hyperparameters used in our experiments (Sec. 4) were selected based on validation loss. For the burn-in epoch $B$, update frequency $F$, and number of steps $K$, we searched over $5, 10, 20, 40$; for the regularization weights $\tau_1$ and $\tau_2$, we used $0.1, 0.5, 1.0$. The graph learning rate $\gamma$ was chosen as $0.1$ or $0.5$. Regarding the prior probabilities, we fixed $\pi^u = 10^{-5}$ and tried $\pi^o$ with $0.99$ and $10^{-4}$. The temperature of the binary concrete distribution (Eq. 9) was set to $0.1$. We optimized $\boldsymbol{\theta}$ with Adam (Kingma & Ba, 2015) and $\phi$ with SGD (Robbins & Monro, 1951), using softplus activations throughout.

Table 5: Hyperparameters of LAGS-GCN for Table 1.

| | Cora | Citeseer | Pubmed | BlogCatalog | Flickr | Roman-Empire |
|---|---|---|---|---|---|---|
| $\gamma$ | 0.1 | 0.5 | 0.1 | 0.1 | 0.1 | 0.5 |
| $\pi^o$ | 0.99 | 0.99 | 0.99 | $1 \times 10^{-4}$ | $1 \times 10^{-4}$ | $1 \times 10^{-4}$ |
| $\pi^u$ | $1 \times 10^{-5}$ | $1 \times 10^{-5}$ | $1 \times 10^{-5}$ | $1 \times 10^{-5}$ | $1 \times 10^{-5}$ | $1 \times 10^{-5}$ |
| $\tau_1$ | 0.5 | 0.5 | 0.1 | 1.0 | 0.0 | 0.0 |
| $\tau_2$ | 1.0 | 1.0 | 1.0 | 0.5 | 0.5 | 0.5 |
| $B$ | 10 | 20 | 20 | 5 | 20 | 20 |
| $F$ | 10 | 10 | 20 | 5 | 10 | 20 |
| $K$ | 10 | 10 | 40 | 10 | 10 | 20 |
| $\eta$ | 0.01 | 0.01 | 0.01 | 0.01 | 0.01 | 0.1 |
| epochs | 400 | 400 | 400 | 200 | 200 | 800 |
| hidden dim. | 64 | 64 | 64 | 64 | 64 | 128 |
| layers | 2 | 2 | 2 | 2 | 2 | 2 |
| w.d. | $5 \times 10^{-3}$ | $5 \times 10^{-2}$ | $5 \times 10^{-4}$ | $5 \times 10^{-7}$ | $5 \times 10^{-7}$ | $5 \times 10^{-5}$ |

## C.5 WILCOXON $p$-VALUES

Tables 8 and 9 provide the Wilcoxon $p$-values for the experiments reported in Tables 1 and 2, respectively. The p-values are calculated between LAGS and the respective baselines (GCN and GraphSAGE) over 10 runs. We observe that LAGS outperforms both GCN and GraphSAGE on

Table 6: Hyperparameters of LAGS-GraphSAGE for Table 1.

|  | Cora | Citeseer | Pubmed | BlogCatalog | Flickr | Roman-Empire |
|---|---|---|---|---|---|---|
| $\gamma$ | 0.5 | 0.5 | 0.5 | 0.5 | 0.5 | 0.1 |
| $\pi^o$ | 0.99 | 0.99 | 0.99 | 0.99 | 0.99 | 0.99 |
| $\pi^u$ | $1 \times 10^{-5}$ | $1 \times 10^{-5}$ | $1 \times 10^{-5}$ | $1 \times 10^{-5}$ | $1 \times 10^{-5}$ | $1 \times 10^{-5}$ |
| $\tau_1$ | 0.5 | 0.1 | 0.1 | 0.5 | 0.1 | 0.1 |
| $\tau_2$ | 1.0 | 1.0 | 0.5 | 1.0 | 0.5 | 0.5 |
| $B$ | 40 | 20 | 40 | 10 | 20 | 20 |
| $F$ | 40 | 10 | 20 | 10 | 20 | 20 |
| $K$ | 20 | 10 | 10 | 20 | 20 | 20 |
| $\eta$ | 0.01 | 0.01 | 0.01 | 0.01 | 0.01 | 0.1 |
| epochs | 400 | 200 | 200 | 200 | 200 | 800 |
| hidden dim. | 64 | 64 | 64 | 64 | 64 | 64 |
| layers | 2 | 2 | 2 | 2 | 2 | 2 |
| w.d. | $5 \times 10^{-3}$ | $5 \times 10^{-2}$ | $5 \times 10^{-4}$ | $5 \times 10^{-7}$ | $5 \times 10^{-7}$ | $5 \times 10^{-5}$ |

Table 7: Hyperparameters of LAGS-GCN* for Table 2.

|  | Cora | Citeseer | Pubmed | Squirrel | Chameleon |
|---|---|---|---|---|---|
| $\gamma$ | 0.5 | 0.5 | 0.1 | 0.5 | 0.5 |
| $\pi^o$ | 0.99 | 0.99 | 0.99 | 0.99 | 0.99 |
| $\pi^u$ | $1 \times 10^{-5}$ | $1 \times 10^{-5}$ | $1 \times 10^{-5}$ | $1 \times 10^{-5}$ | $1 \times 10^{-5}$ |
| $\tau_1$ | 0.1 | 0.1 | 0.1 | 0.1 | 0.1 |
| $\tau_2$ | 1.0 | 1.0 | 0.1 | 1.0 | 0.5 |
| $B$ | 10 | 40 | 5 | 5 | 10 |
| $F$ | 10 | 10 | 5 | 5 | 10 |
| $K$ | 5 | 20 | 10 | 10 | 10 |

most datasets in Table 1 with p-values less than 0.05, indicating statistical significance. In Table 2, LAGS-GCN* also outperforms GCN on Cora, Citeseer, and Chameleon, again with $p$-values below 0.05.

Table 8: Wilcoxon p-values for LAGS-GCN and LAGS-GraphSAGE results in Table 1.

| Dataset | Cora | Citeseer | Pubmed | BlogCatalog | Flickr | Roman-Empire |
|---|---|---|---|---|---|---|
| GCN | 0.0020 | 0.0020 | 0.2324 | 0.0020 | 0.0020 | 0.0020 |
| GraphSAGE | 0.0020 | 0.0371 | 0.1680 | 0.0098 | 0.0137 | 0.0039 |

Table 9: Wilcoxon p-values for LAGS-GCN* results in Table 2.

| Dataset | Cora | Citeseer | Pubmed | Squirrel | Chameleon |
|---|---|---|---|---|---|
| GCN | 0.0020 | 0.0098 | 0.8457 | 0.1601 | 0.0486 |

## C.6 TRAINING STATISTICS

We provide the mean training statistics of LAGS in Table 10. The training time is measured in seconds per iteration, and the GPU memory usage is measured in GB.

Table 10: Training statistics of LAGS-GCN

| Dataset | Cora | Citeseer | Pubmed | Blogcatalog | Flickr | Roman-Empire |
|---|---|---|---|---|---|---|
| Training time per iter (sec) | 0.2675 | 0.4743 | 0.8434 | 2.010 | 4.783 | 1.897 |
| GPU memory usage (GB) | 0.6979 | 1.197 | 31.24 | 3.692 | 7.873 | 42.91 |

## C.7 OTHER GSL METHODS

In Table 1, we include results from other GSL methods. Here, we provide additional details on those methods. Learning Discrete Structures (LDS)(Franceschi et al., 2019) employs bi-level optimization with validation data. Iterative Deep Graph Learning (IDGL)(Chen et al., 2020b) constructs graphs from cosine similarity of node embeddings with sparsity- and smoothness-inducing losses, while IDGL-Anch adopts an anchor-based variant. NodeFormer (Wu et al., 2022) is a graph transformer that applies kernelized Gumbel-Softmax for sparse attention; WSGNN (Lao et al., 2022) employs variational inference with dual branches for observed and learned graphs; SUBLIME (Liu et al., 2022) uses unsupervised bootstrapping contrastive learning with anchor structures; and unGSL (Han et al., 2025) extends a GSL, which we chose to be SUBLIME, by modeling node-level uncertainty. GraphGLOW (Zhao et al., 2023) is a transfer learning framework which trains graph learner on other graph datasets before adapting to the target dataset. We follow the hyperparameters defined in the original papers and the survey of Zhou et al. (2023)

## C.8 ANALYSIS OF THE LEARNED GRAPHS IN LAGS-GCN*

We analyze the properties of the learned graphs from LAGS-GCN* in Table 2 here. Specifically, we analyze the homophily, average degree and average clustering coefficient of the learned graphs. The homophily is the ratio of edges between nodes of the same class to the total number of edges. The average degree is the average number of edges per node, and the average clustering coefficient is a measure of how strongly nodes tend to cluster together. The results in Table 11 include relative increase/decrease from the original graph. We observe a general increase in homophily and cluster coefficient while a decrease in average degree. The increase in homophily aligns with our expectations, as GCNs typically perform better on more homophilic graphs. Similarly, the higher clustering coefficient suggests a shift in node interactions from 2-hop neighborhoods to more localized 1-hop neighborhoods, reflecting GCN's inductive bias toward shorter-range interactions. The reduction in average degree indicates that LAGS tends to remove noisy edges rather than adding new edges.

Table 11: Graph properties of the learned graphs from LAGS-GCN* with the relative increase/decrease from the original graph.

| Dataset | Cora* | Citeseer* | Pubmed* | Squirrel | Chameleon |
|---|---|---|---|---|---|
| homophily | (↑ .10%) 0.8061 | (↑ .29%) 0.7691 | (↑ .20%) 0.7740 | (↑ .18%) 0.6322 | (↑ .17%) 0.5530 |
| avg. degree | (↓ .11%) 1.942 | (↓ .66%) 1.646 | (↓ .04%) 1.878 | (↓ .01%) 6.644 | (↓ .09%) 7.280 |
| avg. cluster coeff. | (↑ .13%) 0.2338 | (↑ .23%) 0.1854 | (↑ .42%) 0.1419 | (↑ .19%) 0.2205 | (↓ .55%) 0.2883 |

# D GRAPH PRIOR

## D.1 ABLATION ON GRAPH PRIOR

Using the same experimental setup described in Section 4 and identical hyperparameters (Appendix C.3), we ablate the effect of the graph prior on LAGS-GCN performance on three configurations. In all configurations, the probability of the observed edge $\pi^o$ is set to 0.99 and the probability of unobserved edges $\pi^u$ is set to 0. The difference between the configurations lies in the use of a k-nearest neighbor (kNN) graph as an additional prior: the first configuration does not use a kNN graph, while the second and third configurations use a kNN graph with $k = 3$ (with probability $\pi^k = 0.1$) and $k = 5$ (with probability $\pi^k = 0.01$) neighbors respectively. Results in Table 12 shows that LAGS can improve performance of GNN even with sparse priors.

Table 12: Node classification results (%) using LAGS-GCN with different graph priors.

| Dataset | Cora | Citeseer | Pubmed | BlogCatalog | Flickr | Roman-Empire |
|---|---|---|---|---|---|---|
| GCN | 81.31 ± 0.46 | 71.19 ± 0.51 | 78.62 ± 0.42 | 75.48 ± 0.37 | 63.71 ± 0.37 | 52.52 ± 0.38 |
| + LAGS case 1 | 82.47 ± 0.24 | 72.94 ± 1.47 | 79.06 ± 0.27 | 75.61 ± 0.38 | 64.21 ± 0.24 | 54.14 ± 0.43 |
| + LAGS case 2 | 82.48 ± 0.42 | 71.30 ± 2.14 | 78.23 ± 0.60 | 76.17 ± 0.38 | 64.98 ± 0.19 | 54.11 ± 0.42 |
| + LAGS case 3 | 82.62 ± 0.38 | 71.84 ± 1.33 | 79.04 ± 0.68 | 75.66 ± 0.60 | 63.85 ± 0.54 | 52.66 ± 0.54 |

## D.2 SCALABILITY

Sparse graph prior (e.g. k-nearest neighbors (kNN) graph) can be used to make learning over large graph structure more tractable. We provide details on the experiment with ogbn-arxiv dataset using GraphSAGE as the base GNN model. The additional hyperparameters $k$ correspond to the number of neighbors in the kNN graph prior and $\pi^k$ is the prior probability of the edges in the kNN graph. We use batch size of 1024 nodes and the total training time took 4.25 hours. The results and training statistics are shown in Table 13 and hyperparameters are in Table 14.

| Dataset | ogbn-arxiv |
|---|---|
| GraphSAGE | $62.51 \pm 0.21$ |
| +LAGS | $\mathbf{63.46} \pm 0.06$ |
| Batch size (nodes) | 1024 |
| Training time per iter (sec) | 38.25 |
| GPU memory usage (GB) | 37.65 |

Table 13: Node classification results (%) and training statistics using the observed graph + $k$-nearest neighbor graph as prior.

| LAGS Hyper | Value | GNN Hyper | Value |
|---|---|---|---|
| $\gamma$ | 0.1 | $\eta$ | 0.01 |
| $\pi^o$ | 0.99 | epochs | 200 |
| $\pi^u$ | 0 | hidden dim. | 64 |
| $k$ | 5 | layers | 2 |
| $\pi^k$ | $1 \times 10^{-5}$ | w.d. | $5 \times 10^{-5}$ |
| $\tau_1$ | 0.0 | | |
| $\tau_2$ | 0.5 | | |
| $B$ | 10 | | |
| $F$ | 10 | | |
| $K$ | 10 | | |

Table 14: Hyperparameters of LAGS-GraphSAGE for ogbn-arxiv (Table 13).

## E VISUALIZATION

Given the inherent difficulty of visualizing even relatively small graphs, we present a summarized visualization (Fig. 5(a)) of the graph structure learned by LAGS-GCN on the Citeseer dataset. Nodes are grouped according to their class labels, with node size proportional to the number of nodes in each label and edge width proportional to the number of edges connecting the corresponding label pairs. There is generally an increase in intra-label edges (black) and a decrease in inter-label edges (red), indicating that LAGS-GCN effectively refines the graph structure to enhance class cohesion. A projection, via principal component analysis (PCA), of the learned final node embeddings (Fig. 5(b)) between GCN and LAGS-GCN show better separation with LAGS-GCN, indicating that the learned graph structure allows it to learn a more discriminative representation of the nodes.

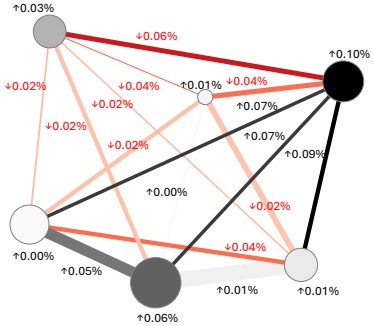
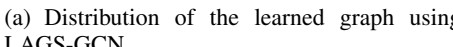
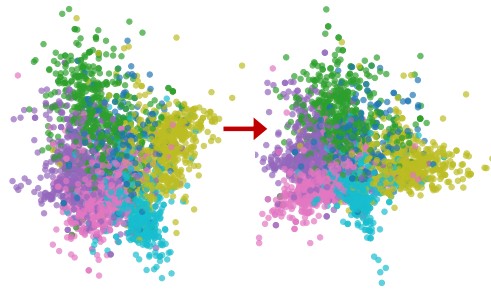

(a) Distribution of the learned graph using LAGS-GCN

(b) Embeddings learned by GCN (*left*) and LAGS-GCN (*right*).

Figure 5: Visualization of Citeseer. (a) distribution of learned graph structure, where node size and edge width indicate counts, and increase/decrease of edges shown in red and black. LAGS-GCN generally increases intra-label and reduces inter-label edges. (b) learned embeddings colored by labels, showing clearer separation and more discriminative representations with LAGS-GCN.

## F  DISCLOSURE STATEMENT ON LLM USAGE

The use of Large Language Models (LLMs) was limited to aiding and polishing writing. No content generation was performed using LLMs.

