# OpenReview forum: "Variational Graph Structure Learning for GNNs by Using Marginal Likelihood"
_ICLR.cc/2026/Conference — Submitted to ICLR 2026_

### Official Review · Reviewer_QBr5 · 2025-10-31

**Soundness:** 3
**Presentation:** 2
**Contribution:** 2
**Rating:** 4
**Confidence:** 4

**Summary:**

This paper proposes the Laplace Approximation-based Graph Structure method to address the challenge of learning optimal graph structures for GNNs. The proposed model leverages marginal likelihood as an objective. It uses a variational formulation with Laplace’s method to derive a marginal likelihood objective over discrete graphs, optimized via the Gumbel-Softmax trick. Empirically, it improves performance across base GNNs (GCN, GraphSAGE).

**Strengths:**

1. Using marginal likelihood avoids rigid, task-specific constraints, enabling automatic regularization that aligns with GNN inductive biases.

2. The variational formulation with Laplace’s method provides a principled link between marginal likelihood and graph structure learning.

3. Ablations show marginal likelihood correlates with generalization and edge importance, providing actionable insights into learned graph quality.

**Weaknesses:**

1. Calculating and inverting the Hessian (even with approximations) increases computational cost. No comparisons of running time were found. For example, on the ogbn dataset, the running time difference between adding and not adding LAGS, and the runtime of other baseline methods.

2. Scalability depends on kNN/observed graph priors, which may exclude potentially optimal edges outside predefined candidates. Whether this part has defects is not supported by quantitative experimental results.

3. Gains on highly homophilic graphs (e.g., Pubmed) are minimal (~0.3%), suggesting limited utility for a large portion of well-structured data.

4. Excessive hyperparameters increase the cost of parameter tuning, raising concerns about its practicality. Furthermore, no sensitivity experimental results on these hyperparameters were provided.

**Questions:**

Please refer to the weakness.

---

> ### Author Response · Authors · 2025-11-20
>
> Thank you for the feedback. Please find below our response:
> - Run time:\
> To clarify the computational cost, we include a runtime comparison with other GSL methods on Cora. LAGS introduces less computational overhead than many alternative GSL approaches.\
> **Table: Runtime (sec/epoch) and GPU memory usage (GB) of LAGS and alternative GSL methods on Cora.**
>
> |                | LAGS-GCN | LAGS-GraphSAGE | WSGNN | NodeFormer | IDGL  | IDGL-Anch | Sublime | Sublime +UnGSL | GraphGlow |
> |----------------|---------|---------------|-------|------------|-------|-----------|---------|---------------|-----------|
> | **Time (s) per epoch** | 0.6185  | 0.3876        | 0.2967 | 0.1041     | 0.3507 | 0.0409    | 1.3704  | 1.4579        | 0.1159    |
> | **GPU (GB)**       | 0.9622  | 0.9714        | 1.844  | 0.6474     | 2.010  | 0.9983    | 0.8263  | 0.9744        | 3.223     |
>
> - Scalability dependence on prior:\
> The computational cost naturally increases with the size of the graph-structure search space, which is determined by the sparsity of the graph prior. A dense prior forces any graph-learning method to consider a large number of candidate edges, leading to substantial memory and computation requirements—a nearly fully connected graph is inherently expensive to learn. Our approach allows direct control over this cost by supporting flexible, compositional priors (e.g. combining the observed graph and kNN graph), since the prior factorizes over edges.
> - Gains on highly homophilic graphs:\
> We believe it is reasonable that when the graph structure is already close to optimal, structure learning yields limited gains. This trend is also seen in other GSL methods on Cora, Citeseer, and Pubmed. Unlike many of these methods—which can even degrade performance—ours still provides some improvements. To further demonstrate the advantage of LAGS, we generate noisy variants of the Cora graph by randomly dropping and adding spurious edges at different rates. The results show that LAGS improves the robustness of the underlying GNN, with accuracy gains of up to 1%.\
> **Table: Accuracy (%) on Cora graphs with artificially dropped or added fractions of edges.**
> | Dataset      | Drop 0.25          | Add 0.25           | Drop 0.50           | Add 0.50            | Drop 0.75          | Add 0.75           |
> |--------------|---------------------|---------------------|---------------------|---------------------|---------------------|---------------------|
> | **GCN**          | 81.75 ± 0.44        | 76.88 ± 0.41        | 79.22 ± 0.41        | 72.56 ± 0.58        | 74.70 ± 1.00        | 70.16 ± 0.71        |
> | **+LAGS**     | **81.99** ± 0.42     | **77.12** ± 0.46     | **80.03** ± 0.35     | **73.22** ± 0.54     | **75.04** ± 0.48     | **71.22** ± 0.70     |
> - Hyperparameters:\
> The graph prior is typically known, for example using the observed graph or a kNN graph. As for the temperatures $(\tau_1, \tau_2)$, they are continuous hyperparameters and can be learned through the same marginal-likelihood objective of the graph structure. We aim to address this in future work.
>
> We are grateful for the reviewer’s feedback, and we would appreciate an updated assessment if the revisions align with the reviewer’s expectations.

---

### Official Review · Reviewer_ZVFA · 2025-11-03

**Soundness:** 2
**Presentation:** 3
**Contribution:** 3
**Rating:** 4
**Confidence:** 3

**Summary:**

This paper proposes LAGS (Laplace Approximation-based Graph Structure Learning), a variational framework for learning graph structures in GNNs via marginal likelihood maximization. The key idea is that the marginal likelihood naturally regularizes the learned structure by balancing model fit and complexity. The paper applies Laplace’s approximation and the Gumbel-Softmax trick to enable optimization over discrete graphs, demonstrating consistent performance gains across homophilic and heterophilic datasets.

**Strengths:**

The paper presents a conceptually clear and theoretically motivated approach connecting Bayesian marginal likelihood to graph structure learning.

The use of Laplace approximation to derive a tractable surrogate objective is technically sound and novel.

Empirical results show consistent improvements over base GNNs.

**Weaknesses:**

The experimental validation is limited in scope relative to the paper’s motivation. Since the paper emphasizes learning from noisy or unreliable graphs, the absence of experiments on synthetic datasets with controlled graph perturbations is a missed opportunity. Such experiments could directly test robustness to varying noise levels and validate the claimed advantage of the method.

Scalability remains a concern, while approximations for the Hessian are discussed, the empirical section does not provide clear runtime or memory comparisons against competing methods.

The empirical gains of stronger GNNs on standard benchmarks, though consistent, are relatively modest (around 1%–2% on many datasets) and may fall within variance bounds.

**Questions:**

How does the proposed method scale with the number of nodes and edges?

---

> ### Author Response · Authors · 2025-11-20
>
> Thank you for the feedback and suggestions. Please see below for our response:
> - Scalability with nodes and edges:\
> Scalability is determined by the number of candidate edges. A fully connected graph (with $N$ nodes) incurs $O(N^2)$, but this cost can be substantially reduced by imposing a sparse prior—such as a kNN graph combined with the observed graph. Under such a prior, computation scales with the number of nonzero edges, giving direct control over the trade-off between efficiency and the size of the graph search space. Moreover, our method supports batching, further improving computational manageability. We refer the reviewer to Section 4.1 (line 371) for additional discussion.
> - Scalability comparison:\
> We provide a runtime comparison with other GSL methods below. As the table shows, LAGS is less computationally intensive than many alternative approaches.\
> **Table: Runtime (sec/epoch) and GPU memory usage (GB) of LAGS and alternative GSL methods on Cora.**
>
> |                | LAGS-GCN | LAGS-GraphSAGE | WSGNN | NodeFormer | IDGL  | IDGL-Anch | Sublime | Sublime +UnGSL | GraphGlow |
> |----------------|---------|---------------|-------|------------|-------|-----------|---------|---------------|-----------|
> | **Time (s) per epoch** | 0.6185  | 0.3876        | 0.2967 | 0.1041     | 0.3507 | 0.0409    | 1.3704  | 1.4579        | 0.1159    |
> | **GPU (GB)**       | 0.9622  | 0.9714        | 1.844  | 0.6474     | 2.010  | 0.9983    | 0.8263  | 0.9744        | 3.223     |
>
>
> - Significance of the empirical gain:\
> We report the Wilcoxon p-values in Appendix C.5 (reproduced below) to demonstrate statistical significance. Most p-values fall below 0.05, confirming that our improvements are statistically significant.\
> **Table: Wilcoxon p-values of results in Table 1 (paper)**
> | Dataset     | Cora   | Citeseer | Pubmed  | BlogCatalog | Flickr | Roman-Empire |
> |-------------|--------|----------|---------|-------------|--------|--------------|
> | **LAGS-GCN**     | 0.0020 | 0.0020   | 0.2324  | 0.0020      | 0.0020 | 0.0020       |
> | **LAGS-GraphSAGE** | 0.0020 | 0.0371   | 0.1680  | 0.0098      | 0.0137 | 0.0039       |
> - Experiments on synthetic data:\
> Following the reviewer’s suggestion,  we conducted additional experiments on Cora by artificially introducing noise through random edge deletions and insertions. For edge deletion, we randomly removed edges with probabilities 0.25, 0.50, and 0.75. For edge addition, we randomly inserted new edges at ratios of 0.25, 0.50, and 0.75 relative to the original number of edges. We evaluated both the baseline GCN and LAGS-GCN over 10 runs. Across all settings, LAGS improves the robustness of GCN under noisy conditions—particularly under noisy edge insertions—with more than 1% accuracy gain.\
> **Table: Accuracy (%) on Cora graphs with artificially dropped or added fractions of edges.**
> | Dataset      | Drop 0.25          | Add 0.25           | Drop 0.50           | Add 0.50            | Drop 0.75          | Add 0.75           |
> |--------------|---------------------|---------------------|---------------------|---------------------|---------------------|---------------------|
> | **GCN**          | 81.75 ± 0.44        | 76.88 ± 0.41        | 79.22 ± 0.41        | 72.56 ± 0.58        | 74.70 ± 1.00        | 70.16 ± 0.71        |
> | **+LAGS**     | **81.99** ± 0.42     | **77.12** ± 0.46     | **80.03** ± 0.35     | **73.22** ± 0.54     | **75.04** ± 0.48     | **71.22** ± 0.70     |
>
> If our revisions and clarifications have successfully addressed the reviewer’s concerns, we would appreciate it if the reviewer could update the evaluation/score.

---

### Official Review · Reviewer_pqiU · 2025-11-05

**Soundness:** 1
**Presentation:** 2
**Contribution:** 2
**Rating:** 2
**Confidence:** 4

**Summary:**

Accurately learning graph structures is fundamental, as it enables numerous downstream applications and provides substantial benefits for real-world scenarios.

**Strengths:**

Accurately learning graph structures is fundamental, as it enables numerous downstream applications and provides substantial benefits for real-world scenarios.

**Weaknesses:**

1. The proposed model is tested only on node classification task, however several types of graph based task exists and are not considered in the paper. This raises a natural question if the approach is limited to node classification and cannot be generalized to other graph based tasks like link prediction, and graph classification?
2. The result table shows the effect of proposed method, LAGS, on GraphSAGE and GCN. The gain is really low compared to GraphSAGE nullifying the contribution from LAGS. Moreover compared  to the other GSL (Graph Structure Learning)  methods like LDS and SUBLIME methods, there is no visible advantage of using LAGS.
3. The formatting in Table 1 is not consistent as it does no explain the meaning of bold face as bold face generally indicate the highest score.
4. Overall the evaluation is weak.

**Questions:**

see above

---

> ### Author Response · Authors · 2025-11-20
>
> Thank you for your feedback. Please see below for our response:
> - Limitation to node classification:\
> Our method learns a parameter for every potential edge and therefore requires a fixed graph, making it unsuitable for graph-level tasks where the graph varies across inputs. This is common across most graph structure learning (GSL) methods (e.g. LDS, WSGNN, SUBLIME, etc.), which are primarily designed for node-level tasks. GSL tends to yield the largest gains in these settings because predictions depend heavily on local neighborhoods, and noisy or missing edges—common in citation, social, and hyperlink networks—directly harm node representations. In contrast, graph-level tasks rely on pooled representations and typically operate on clean, well-defined structures such as molecular graphs, where structure learning offers limited benefit.
> Following the reviewer’s suggestion, we conducted additional link prediction experiments with LAGS. Although structure learning is not typically used for link prediction—since the task itself infers edges—we find that LAGS remains effective. By learning a more informative supporting graph, LAGS improves the quality of the node embeddings and, consequently, the downstream edge predictions.\
> **Table: Link prediction accuracy (%) across datasets.**
> | Dataset    | Cora            | Citeseer        | BlogCatalog      |
> |------------|------------------|------------------|-------------------|
> | **GCN**        | 59.67 ± 0.82     | 57.84 ± 1.11     | 76.09 ± 0.30      |
> | **+LAGS**   | **60.18** ± 0.66     | **58.54** ± 0.88     | **76.77** ± 0.28      |
>
> - Significance of results:\
> We report the Wilcoxon p-values for our experimental results in Appendix C.5 (also reproduced below). Most p-values fall below 0.05, confirming that our improvements on GCN and GraphSAGE are statistically significant.\
> **Table: Wilcoxon p-values of results in Table 1 (paper)**
> | Dataset     | Cora   | Citeseer | Pubmed  | BlogCatalog | Flickr | Roman-Empire |
> |-------------|--------|----------|---------|-------------|--------|--------------|
> | **LAGS-GCN**     | 0.0020 | 0.0020   | 0.2324  | 0.0020      | 0.0020 | 0.0020       |
> | **LAGS-GraphSAGE** | 0.0020 | 0.0371   | 0.1680  | 0.0098      | 0.0137 | 0.0039       |
>
> Our approach does not modify the underlying GNN architecture, unlike methods such as NodeFormer, WSGNN, and SUBLIME. As a result, the performance gains depend on how strongly the underlying GNN relies on the graph structure (e.g., larger gains for GCN than for GraphSAGE), but because we do not modify the base GNN, our method is easily adaptable to new architectures, allowing it to benefit from future advances in GNN design.\
> Regarding comparison with other GSL methods, LDS and IDGL are far more computationally intensive—unable to finish on several benchmarks—and no other method consistently outperforms LAGS. We further discuss why our method is not directly comparable to the alternatives below.
> - Formatting in Table 1:\
> LAGS is directly comparable only to the baseline GNNs. It is not directly comparable to many existing GSL methods because those approaches substantially alter the GNN architecture (e.g., NodeFormer replaces the GNN with a Transformer backbone; WSGNN uses two GNNs; SUBLIME introduces auxiliary learning modules), rely on another GSL algorithm as a prerequisite (unGSL), or require additional data sources (GraphGLOW uses transfer learning; LDS uses validation labels during graph learning). Additional details are provided in Appendix C.7. We attempted to clarify this in lines 364–370, but we understand that the original phrasing and table may have been confusing. We will revise the text to emphasize this distinction more explicitly.
> - Evaluation:\
> To further demonstrate the advantage of LAGS, we show here the benefit that LAGS bring to noisy graphs. Using Cora, we randomly removed edges at different probabilities and injected spurious edges at varying ratios relative to the original edge count. The result below shows the gains that LAGS bring especially when noisy edges are injected, with performance gain of more than 1%.\
> **Table: Accuracy (%) on Cora graphs with artificially dropped or added fractions of edges.**
> | Dataset      | Drop 0.25          | Add 0.25           | Drop 0.50           | Add 0.50            | Drop 0.75          | Add 0.75           |
> |--------------|---------------------|---------------------|---------------------|---------------------|---------------------|---------------------|
> | **GCN**          | 81.75 ± 0.44        | 76.88 ± 0.41        | 79.22 ± 0.41        | 72.56 ± 0.58        | 74.70 ± 1.00        | 70.16 ± 0.71        |
> | **+LAGS**     | **81.99** ± 0.42     | **77.12** ± 0.46     | **80.03** ± 0.35     | **73.22** ± 0.54     | **75.04** ± 0.48     | **71.22** ± 0.70     |
>
> We would be thankful if the reviewer could consider updating the evaluation, provided that our responses have addressed the issues raised.

---

### Meta-Review · Area_Chair_om9W · 2025-12-22

**Summary:**

The reviewers expressed concerns that the methods mentioned in the paper are only limited to node classification tasks, the computational complexity, the limited improvement in model performance, and the testing in more robust scenarios. Although the author made some responses, on the whole, the advantages of the proposed model were not obvious.

**Reviewer Concerns:**

I don't think the author's response can address the reviewers' concerns.
1. Limited to node classification task: The author supplemented the experiments on link prediction. However, the experimental results of GCN are strange and far from the general link prediction performance. The author needs to be more clear about the experimental setup.
2. Computational complexity: The author supplemented relevant experiments, but there is still a gap in performance compared with some methods, although it surpasses many methods. In addition, theoretically, this method still has the problem of considerable complexity.
3. Limited Improvement and more robust scenarios: The author's improvement in the synthetic dataset is still very limited, which is worrying.

**Reviewer Scores:**

I think the Reviewer pqiU might be raised to 4 points or remain unchanged. The other two reviewers would not consider raising their scores because the authors' responses were not convincing enough. In any case, the final score of this article may still be in a rejected position.

---

### Decision · Program_Chairs · 2026-01-26

Reject